# *Fusobacterium nucleatum* promotes tumor extravasation and metastasis in head and neck cancer via TLR4/MYB/ESPN axis

Xiaohui Yuan [1,2,3], Huiying Huang[1,2,3], Zhenwei Wang[1,2], Hui-Ching Lau[1,2], Qiang Huang [1,2], Yujie Shen[1,2], Ming Zhang[1,2], Lei Tao[1,2], Hongli Gong [1,2,4] ✉, Chi-Yao Hsueh [1,2,4] ✉ & Liang Zhou [1,2,4] ✉

Patients with metastasis have an extremely poor prognosis in head and neck squamous cell carcinoma (HNSCC). Emerging studies have illuminated the impact of intratumor microbiota on cancer metastasis, though the specific role of *Fusobacterium nucleatum* (*F. nucleatum*) in HNSCC metastasis remains unresolved. This research found that intratumoral *F. nucleatum* abundance was elevated and correlated to diminished disease-free survival in HNSCC patients exhibiting lymph node metastasis. *F. nucleatum* invasion into primary and metastatic tumor tissues was observed using fluorescence in situ hybridization. *F. nucleatum* induced adhesion to endothelial cells and facilitated transendothelial migration via upregulating ESPN expression in HNSCC cells, which is an actin-binding protein. Mechanistically, *F. nucleatum* activated TLR4 signaling, inducing elevated expression of the transcription factor MYB, which subsequently stimulated ESPN transcription. Furthermore, metronidazole treatment significantly reduced metastatic incidence in vivo. These results indicate the significant potential of targeting *F. nucleatum* as a therapeutic approach in metastatic HNSCC.

Head and neck squamous cell carcinoma (HNSCC) is a highly prevalent malignant tumor worldwide[1]. Patients with HNSCC endure significant impairments in swallowing, speaking, and breathing, severely compromising their quality of life. More than 50% of HNSCC cases are diagnosed at advanced stages or with lymph node metastasis (LNM) due to the tumor's aggressive invasion and metastatic properties[2,3]. Despite ongoing advancements and refinements in treatment modalities, long-term survival rates remain stagnant[4]. Therefore, addressing HNSCC metastasis is crucial to improving patients' outcomes, necessitating the identification of risk factors and molecular mechanisms that drive metastasis for the development of targeted, effective therapies.

The HNSCC tumorigenesis results from a complex interplay between genetic factors and environmental exposures, showing considerable levels of heterogeneity[5]. In addition to the primary risk factors for HNSCC—tobacco use, alcohol consumption, and human papillomavirus (HPV) infection—emerging attention has been directed toward the potential role of microbiota in HNSCC development. A large study revealed that heavy drinkers exhibit significant changes in their oral microbial structure, showing the influence of alcohol on the oral microbiome[6]. Notably, the abundance of specific bacteria increases under the influence of alcohol, and their metabolites—which may serve as a source of carcinogenic acetaldehyde—could play a role in mediating the carcinogenic effects of alcohol consumption[7]. HPV-positive HNSCC exhibits alterations in the abundance of specific bacteria[8] and on the other hand, certain oral microbiota structure may increase host susceptibility to HPV infection and potentiate its cancer-promoting effects. The human microbiota has gained increasing recognition as a key player in cancer development and progression[9].

In the past two decades, the human microbiota has gained increasing recognition as a key player in cancer development and progression. Advances in microbiome research have opened new avenues for cancer treatment and investigation[10]. *Fusobacterium nucleatum* (*F. nucleatum*) is a gram-negative anaerobic bacterium predominantly found in the oral cavity and gut. Its relationship with tumor metastasis has received increasing attention[11–13]. A recent study revealed that *F. nucleatum* strains isolated from both primary and metastatic lesions in the same patient exhibited similar DNA, suggesting its involvement in metastasis[14]. Previous researches have

[1]ENT institute and Department of Otorhinolaryngology, Eye & ENT Hospital, Fudan University, Shanghai, PR China. [2]Shanghai Key Clinical Disciplines of Otorhinolaryngology, Shanghai, PR China. [3]These authors contributed equally: Xiaohui Yuan, Huiying Huang. [4]These authors jointly supervised this work: Hongli Gong, Chi-Yao Hsueh, Liang Zhou. ✉e-mail: gonghlent@126.com; hsuehchiyao@gmail.com; zhoulent@126.com

demonstrated the link between *F. nucleatum* and tumor progression, identifying it as a prognostic risk factor for laryngeal and hypopharyngeal cancers[15,16]. Using xenograft mouse models, previous study established that *F. nucleatum* enhances the progression and epithelial-mesenchymal transition (EMT) via TLR4/MYD88/TGFβR2 pathway in laryngeal cancer[17]. However, despite accumulating evidence, the specific role and mechanisms through which *F. nucleatum* influences HNSCC metastasis remain unclear.

Metastasis is a complex and multi-stage process involving migration, invasion, adhesion, and extravasation[18]. The process of extravasation involves tumor cell adhesion to endothelial cells (ECs), followed by trans-endothelial migration. Moreover, tumor cells use filopodia to locate optimal sites for transendothelial migration[19]. Actin cytoskeleton remodeling is a vital step as tumor cells traverse the narrow spaces between ECs. ESPN, an actin-binding protein encoded by the *espin* gene, regulates this process. Its WH2 domain binds to actin monomers, promoting the bundling of actin filaments[20]. ESPN is crucial for cytoskeletal remodeling, especially in the formation and extension of cellular protrusions like filopodia and stereocilia[21], highlighting its role in facilitating tumor migration and invasion in cancers. ESPN is highly expressed in melanoma and knockdown of ESPN expression resulted in reduced lamellipodia formation as well as decreased activities of FAK and Rho GTPases in melanoma cells[22]. In a recent research, ESPN was reported to promote osteochondroma metastasis. ESPN activated the AKT/mTOR pathway via direct interaction with and phosphorylation of PI3K, which enhanced ZEB1 expression to initiate EMT[23]. High expression of ESPN is associated to poor prognosis in esophageal squamous cell carcinoma[24]. However, the role of ESPN in promoting metastasis in squamous cell carcinoma, particularly in HNSCC, remains unclear.

This study aims to clarify the effects of *F. nucleatum* on HNSCC metastasis, along with its underlying molecular mechanisms. Our findings demonstrate that *F. nucleatum* increases HNSCC cells adhesion to ECs and enhances the abilities to undergo transendothelial migration, promoting tumor extravasation. In vivo experiments further indicated that antibiotic treatment mitigated *F. nucleatum*-induced metastasis. Mechanistically, *F. nucleatum* modulates the interaction between HNSCC cells and ECs through the TLR4/ESPN/MYB axis.

## Results

### *F. nucleatum* participates in lymph node metastasis and is an independent prognostic factor in HNSCC

Mounting evidence demonstrated that *F. nucleatum* is crucial in tumor metastasis and is associated with poor clinical outcomes[14,25]. To clarify the clinical significance of *F. nucleatum* in HNSCC, we collected five paired primary tumor tissues and metastatic lymph node tissues. As shown in Fig. 1A, *F. nucleatum* was detected by fluorescence in situ hybridization (FISH) in both primary and metastatic tissues from three patients, suggesting its involvement during HNSCC metastasis. To further elucidate the association between *F. nucleatum* and LNM, we constructed a tissue microarray (TMA) using fresh HNSCC tumor tissues (Cohort 1, *n* = 77) and collected formalin-fixed paraffin-embedded (FFPE) HNSCC samples (Cohort 2, *n* = 104). Clinical data are detailed in Table 1. *F. nucleatum* levels in HNSCC tissues (Cohort 1) were quantified using FISH. The results revealed a significant increase in *F. nucleatum* levels from patients with LNM (LNM group) compared to those without LNM (NLNM group) (Fig. 1B, C). Further, genomic DNA from FFPE HNSCC samples (Cohort 2) was extracted, and *F. nucleatum* abundance was measured using RT-qPCR. The results showed that, compared to the NLNM group, there were higher *F. nucleatum* levels in the LNM group (Fig. 1E). Additionally, *F. nucleatum* levels were elevated in advanced tumors (Fig. 1D, F). Similar results by TCMA cohort are also illustrated in Supplementary Fig. S1A. However, the association between *F. nucleatum* abundance and LNM were not found in the TCMA cohort (Supplementary Fig. S1B). Based on the median abundance of *F. nucleatum*, patients were stratified into two groups: a *F. nucleatum*-high group and a *F. nucleatum*-low group. Kaplan-Meier analysis revealed a significantly shorter disease-free survival (DFS) time in patients

with high *F. nucleatum* levels (Fig. 1G). Given an a priori sample size calculation was not performed, we performed a post-hoc power analysis. Based on the observed hazard ratio of 2.207, actual sample size of 104, and a two-sided α of 0.05, the achieved power for detecting the observed effect was calculated to be 0.874. Furthermore, high abundance of *F. nucleatum* was identified as an independent risk factor for DFS by multivariate Cox regression analysis (Fig. 1H). These results indicated that high abundance of *F. nucleatum* is associated with LNM and poorer prognosis in HNSCC.

### *F. nucleatum* enhances HNSCC cells extravasation through modulating tumor cell adhesion to the endothelium and trans-endothelial migration

Emerging studies have uncovered the biological functions of intratumor microbiota in the metastatic process, with *F. nucleatum* playing a significant role[26]. To assess whether *F. nucleatum* influences HNSCC metastasis, AMC-HN-8 cells were intravenously injected into mice after being incubated with *F. nucleatum* for 24 h. Intravital imaging at 4 weeks revealed tumor cell dissemination to distant sites, with no notable differences in signal intensity between the two groups (Fig. 2A). However, by 8 weeks, the *F. nucleatum*-treated group exhibited metastatic foci with significantly increased signal intensity (Fig. 2B). Mice were sacrificed immediately following imaging, and metastatic foci were identified based on the locations indicated by imaging (Fig. 2C). H&E staining confirmed that the incidence of metastasis increased in *F. nucleatum*-treated group (Fig. 2D). The results indicated that *F. nucleatum* promotes the metastatic capacity of HNSCC cells in vivo.

To further elucidate the pro-metastatic effect of *F. nucleatum*, transwell assays were performed. Migration and invasion assays found that invasive and migratory abilities of AMC-HN-8 and FaDu cells markedly increased after incubating with *F. nucleatum* (Fig. 2E, F). During metastasis, tumor cell extravasation requires direct interaction with endothelium, which involves tumor cell adhesion to the endothelium and subsequent transendothelial migration. To explore the effect of *F. nucleatum* on HNSCC cell extravasation, cancer endothelium adhesion and transendothelial migration assays were conducted[27]. The cancer endothelium adhesion assay showed that the number of adhering AMC-HN-8 and FaDu cells to ECs increased after incubation with *F. nucleatum* (Fig. 2G). Additionally, the *F. nucleatum*-treated group showed a significant increase in the number of tumor cells which passed a single endothelial cell layer (Fig. 2H). These findings collectively indicate that *F. nucleatum* promotes the abilities of tumor cell adhesion to the endothelium and transendothelial migration to enhance the extravasation of HNSCC cells during metastasis.

### *F. nucleatum* promotes HNSCC cells extravasation by upregulating ESPN protein

We next sought to explore the mechanism through which *F. nucleatum* promotes the extravasation of HNSCC cells. The dynamic reorganization of the cytoskeleton is intricately linked to the metastasis of malignant cells[28]. To evaluate the influence of *F. nucleatum* on the cytoskeleton of HNSCC cells, actin cytoskeleton staining was performed after incubating AMC-HN-8 and FaDu cells with *F. nucleatum* for 24 hours. Fluorescent staining revealed that *F. nucleatum* caused actin cytoskeleton disorganization and increased the formation of filopodia (Fig. 3A). To investigate how *F. nucleatum* influences actin cytoskeleton of HNSCC cells, RNA sequencing was conducted to analyze and compare gene expression profiles between *F. nucleatum*-treated and untreated AMC-HN-8 cells. The RNA sequencing data identified 267 differentially expressed genes (DEGs) in the *F. nucleatum*-treated group, comprising 223 upregulated and 44 downregulated genes (Fig. 3B). Among these DEGs, ESPN emerged as a key candidate. ESPN, an actin-binding protein, has been reported to bind F-actin, remodel the cytoskeleton, and regulate the formation of cellular protrusions[29,30].

To validate the RNA sequencing results, the ESPN expression was assessed. The findings indicated that *F. nucleatum* caused a significant time-dependent increase in ESPN expression at both the mRNA and protein levels (Fig. 3C, D). Immunofluorescence staining further revealed that ESPN localized predominantly in the cytoplasm and was notably elevated in *F.*

*nucleatum*-treated HNSCC cells (Fig. 3E). To investigate the functional role of ESPN, its expression was reduced via RNA interference (Supplementary Fig. S2A). Silencing ESPN significantly inhibited the invasive and migratory capacities of AMC-HN-8 and FaDu cells, as well as reduced their adhesion to ECs and transendothelial migration (Fig. 3F, G; Supplementary Fig. S2B, C). Moreover, silencing ESPN markedly attenuated the pro-invasive and metastatic effects induced by *F. nucleatum* (Supplementary Fig. S2B, C). Notably, silencing ESPN also reversed the enhanced tumor cell adhesion to ECs and transendothelial migration stimulated by *F. nucleatum* (Fig. 3F, G). Collectively, these findings indicate that *F. nucleatum* facilitates the interaction between HNSCC cells and ECs via an ESPN-dependent mechanism.

### *F. nucleatum* upregulates MYB to elicit ESPN transcription

To elucidate the regulatory mechanism of *F. nucleatum* on ESPN gene expression, transcription factors predicted to bind to the ESPN promoter region were identified using the JASPAR, CIS-BP, hTFtarget, and HOCOMOCO databases (Supplementary Data 2). After integrating RNA sequencing data, MYB was identified as a candidate regulator (Fig. 4A). RT-qPCR and Western blot analyses indicated that *F. nucleatum* induced a significant time-dependent upregulation in MYB expression at both the mRNA and protein levels (Fig. 4B, C). To investigate MYB's role in regulating ESPN, MYB expression was silenced using siRNAs (Supplementary Fig. S3A), which resulted in decreased ESPN levels and attenuated the *F. nucleatum*-induced upregulation of ESPN (Fig. 4D), suggesting that ESPN is likely a transcriptional target of MYB.

To confirm MYB's direct binding to ESPN promoter region and transcriptional activation, three ESPN promoter fragments (P1: -2,000 to +99, P2: -1000 to +99, and P3: -500 to +99) were cloned into a luciferase reporter plasmid. All three fragments activated the luciferase reporter, with the MYB-mediated transcriptional activation primarily occurring through the P3 fragment (Fig. 4E). The JASPAR and hTFtarget databases were

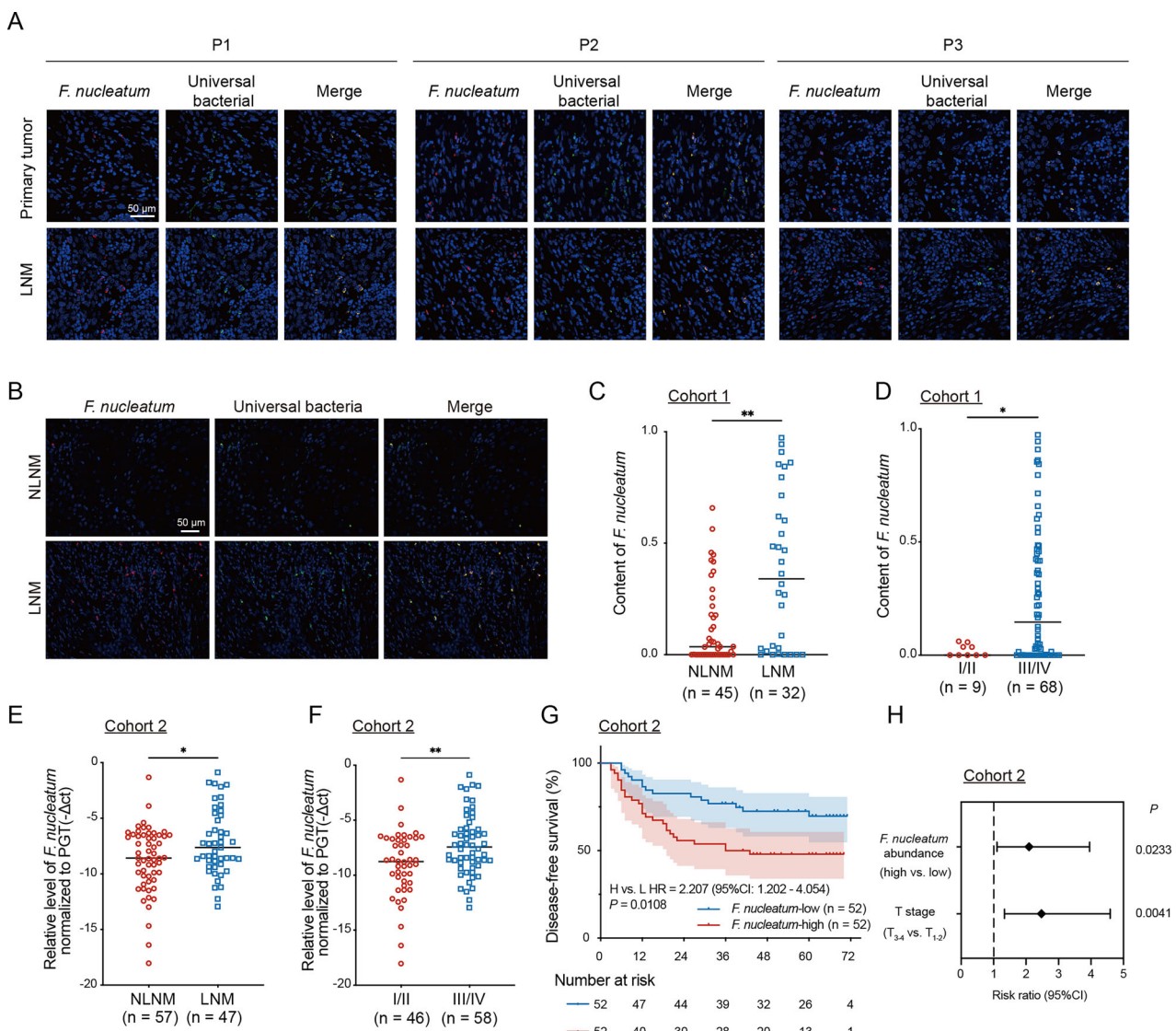

**Fig. 1 | *F. nucleatum* participates in lymph node metastasis and is an independent prognostic factor in HNSCC. A** Detection of *F. nucleatum* in both primary and paired metastatic tumor samples using FISH (*n* = 5). **B** Representative FISH images showing *F. nucleatum* in HNSCC tissues from Cohort 1. **C** Relative abundance of *F. nucleatum* in patients with HNSCC with or without lymph node metastasis in Cohort 1. **D** Relative abundance of *F. nucleatum* in patients with HNSCC across different TNM stages in Cohort 1. **E** Relative abundance of *F. nucleatum* in patients with HNSCC with or without lymph node metastasis in Cohort 2. **F** Relative abundance of *F. nucleatum* in patients with HNSCC across different TNM stages in Cohort 2. **G** Kaplan-Meier survival curve of disease-free survival based on *F. nucleatum* abundance in Cohort 2. **H** Multivariate Cox regression analysis of disease-free survival in Cohort 2. The *F. nucleatum*-specific probe is labeled with CY-3 (red), and the EUB338 universal bacterial probe is labeled with 488 (green). Bars indicate median values. *$P < 0.05$, **$P < 0.01$. LNM, lymph node metastasis. NLNM, no lymph node metastasis.

**Table 1 | Clinicopathological characteristics of HNSCC patients enrolled in this study**

| Variables | Cohort 1, n = 77 (%) | Cohort 2, n = 104 (%) | Cohort 3, n = 78 (%) |
|---|---|---|---|
| Age | | | |
| ≤60 | 20 (26.0) | 51 (49.0) | 37 (47.4) |
| >60 | 57 (74.0) | 53 (51.0) | 41 (52.6) |
| Sex | | | |
| Male | 75 (97.4) | 102 (98.1) | 77 (98.7) |
| Female | 2 (2.6) | 2 (1.9) | 1 (1.3) |
| Smoking history | | | |
| Yes | 65 (84.4) | 82 (78.8) | 65 (83.3) |
| No | 12 (15.6) | 22 (21.2) | 12 (15.4) |
| Drinking history | | | |
| Yes | 43 (55.8) | 46 (44.2) | 45 (57.7) |
| No | 34 (44.2) | 58 (55.8) | 33 (42.3) |
| T stage[a] | | | |
| T1 | 1 (1.3) | 14 (13.5) | 11 (14.1) |
| T2 | 20 (26.0) | 54 (51.9) | 40 (51.3) |
| T3 | 31 (40.3) | 25 (24.0) | 21 (26.9) |
| T4 | 25 (32.5) | 11 (10.6) | 6 (7.7) |
| N stage[a] | | | |
| N0 | 45 (58.4) | 57 (54.8) | 43 (55.1) |
| N1 | 8 (10.4) | 14 (13.5) | 7 (9.0) |
| N2 | 20 (26.0) | 22 (21.2) | 18 (23.1) |
| N3 | 4 (5.2) | 11 (10.6) | 10 (12.8) |
| TNM stage[a] | | | |
| Stage I-II | 9 (11.7) | 46 (44.2) | 38 (48.7) |
| Stage III-IV | 68 (88.3) | 58 (55.8) | 40 (51.3) |

[a]TNM stage was based on the eighth edition of the AJCC.

utilized to predict potential MYB binding sites within the P3 fragment of the ESPN promoter. The highest-scoring site (WT1) was selected from the JASPAR database, while the most frequently repeated site (WT2) was chosen from the hTFtarget database (Supplementary Table S1 and S2, Supplementary Fig. S3B). Mutations introduced at WT1 (1822, 1831) and WT2 (1887, 1968) partially reduced the MYB-induced activation of the ESPN promoter, as shown by promoter reporter assays (Fig. 4F). Then, we performed chromatin immunoprecipitation (ChIP) and electrophoretic mobility shift assay (EMSA) to determine the direct binding of MYB and the ESPN promoter. ChIP-qPCR revealed significant ESPN enrichment in the anti-MYB group compared to the IgG control, confirming that MYB binds to the ESPN promoter (Fig. 4G). Additionally, EMSA demonstrated MYB's binding capacity to the WT2 site in the ESPN promoter (Fig. 4H). Collectively, these results confirm that MYB directly binds to the ESPN promoter to activate its transcription.

*F. nucleatum* is known to influence TLR4 activation in various cancers, and our prior study revealed the activation of TLR4/MYD88 signaling pathway by *F. nucleatum* in AMC-HN-8 cells[17,31]. Thus, it was hypothesized that *F. nucleatum* upregulates MYB via TLR4. As expected, silencing TLR4 reduced MYB levels and attenuated the *F. nucleatum*-induced upregulation of MYB (Fig. 4I). Collectively, these results confirm that *F. nucleatum* activates MYB via TLR4, leading to the transcriptional regulation of ESPN.

**ESPN and MYB levels are linked to *F. nucleatum* abundance and elevated in patients with HNSCC exhibiting LNM**

We next further investigate the clinical significance of ESPN and MYB levels. The clinical association between *F. nucleatum* and ESPN expression was investigated. FISH analysis was conducted to quantify ESPN levels in

the HNSCC tissue microarray (Cohort 1), with samples categorized into *F. nucleatum*-high (n = 38) and *F. nucleatum*-low (n = 39) groups based on the median *F. nucleatum* levels in Cohort 1. Notably, ESPN expression was elevated in the *F. nucleatum*-high group (Fig. 5A). A positive correlation was also observed between *F. nucleatum* levels and ESPN expression (Fig. 5B). To further assess the clinical significance of ESPN, its expression was analyzed in relation to LNM. ESPN levels were significantly increased in HNSCC tissues with LNM (Fig. 5C) a finding consistent with results from the Cancer Genome Atlas (TCGA) cohort (Fig. 5D). Given previous findings that MYB regulates ESPN expression, we explore the correlation between ESPN and MYB. The analysis of the TCGA-HNSC dataset demonstrated a positive correlation between ESPN and MYB mRNA expression (Fig. 5E). Co-expression of MYB and ESPN in HNSCC tissues from Cohort 1 was examined. Figure 5F shows co-localization of ESPN with MYB in regions colonized by *F. nucleatum*. To further validate the clinical relevance of ESPN and MYB, 79 FFPE HNSCC samples were analyzed (Cohort 3), and immunohistochemistry (IHC) results revealed significantly higher expression of ESPN and MYB in the LNM group (Fig. 5G, H). Furthermore, the expression level of ESPN in HNSCC samples was validated using the GEO database (Supplementary Fig. S1C, D). These results collectively indicate a significant positive correlation between the over-abundance of *F. nucleatum* and increased ESPN and MYB levels in HNSCC tissue with LNM.

**Metronidazole treatment reduces *F. nucleatum*-mediated metastasis of HNSCC cells in vivo**

Since metronidazole (MTZ) has been shown to reduce *F. nucleatum* load and effectively inhibit tumor metastasis in various mouse models[12,14], the potential of MTZ to suppress the pro-metastatic effects of *F. nucleatum* on HNSCC cells in vivo was evaluated. Mice were allocated into three groups: Control, *F. nucleatum*, and *F. nucleatum* + MTZ. AMC-HN-8 cells were intravenously injected after 24 h incubation with either *F. nucleatum* or PBS. MTZ (100 mg/kg/day) or saline was administered daily via oral gavage beginning the day after cell injection, continuing for two weeks (Fig. 6A). Intravital imaging at 4 and 8 weeks assessed the intensity and location of metastatic foci. Results revealed that the metastatic foci in the *F. nucleatum* group exhibited higher signal intensity compared with the *F. nucleatum* + MTZ group (Fig. 6B, C). Dissection and H&E staining showed a significant increase in both the incidence of metastasis and the size of metastatic foci in the *F. nucleatum* group (Fig. 6D–F). These results demonstrate that MTZ treatment significantly reduces *F. nucleatum*-mediated metastasis in vivo.

**Discussion**

The involvement of *F. nucleatum* in cancer metastasis has gained increasing attention. In this study, higher *F. nucleatum* levels were detected in tumor tissues of patients with HNSCC exhibiting LNM and correlated with poor prognosis. Furthermore, *F. nucleatum* was found to enhance HNSCC cell adhesion to vascular endothelium, promoting transendothelial migration and thereby facilitating extravasation and promoting metastasis. Mechanistically, *F. nucleatum* activated TLR4/MYB axis, which increased ESPN expression at the transcriptional level.

This study revealed that *F. nucleatum* promotes extravasation and metastasis of HNSCC cells, which aligning with recent studies that demonstrate *F. nucleatum*'s role in enhancing metastasis across various cancers[12,13,32]. More importantly, clinical sample analysis verified that *F. nucleatum* abundance is associated with LNM. However, the TCMA cohort did not reveal a significant difference in *F. nucleatum* abundance regarding LNM. In contrast, a study regarding oral carcinoma reported a lower incidence of LNM in *F. nucleatum*-positive tumor tissues[33]. The inherent heterogeneity of HNSCC must be considered. Tumors originating from different anatomical sites exhibit distinct molecular characterization and tumor behaviors[34]. The oncogenic effects of *F. nucleatum* may demonstrate significant heterogeneity depending on the specific anatomical site and molecular subtype context in which it acts. In colorectal cancer (CRC), *F.*

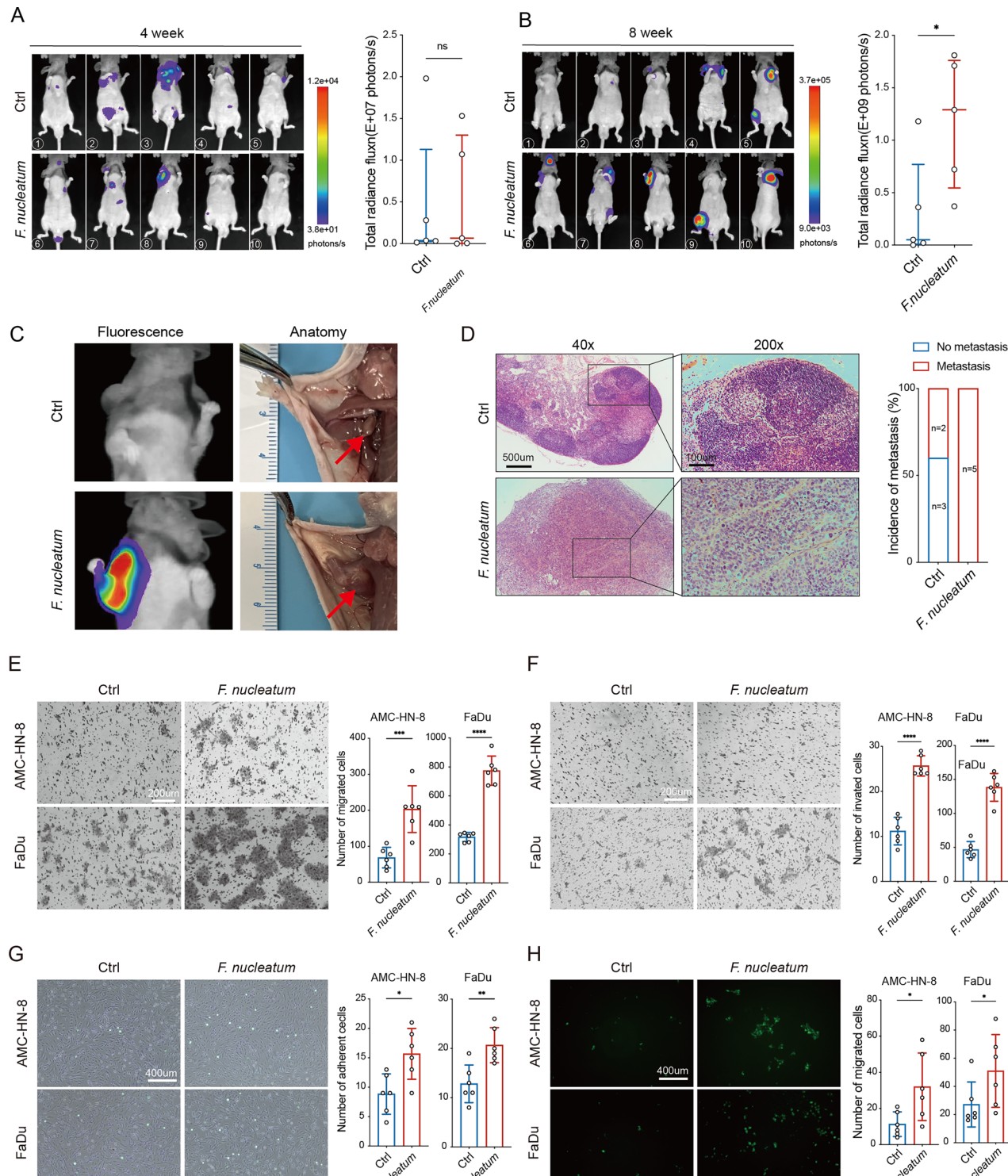

**Fig. 2 | *F. nucleatum* enhances HNSCC cells extravasation through modulating tumor cell adhesion to the endothelium and transendothelial migration.**
**A**, **B** After incubated with *F. nucleatum* or PBS, AMC-HN-8 cells were intravenously injected into mice via the tail vein, and metastatic foci were detected using intravital imaging (*n* = 5 per group). Signal intensity of the metastatic foci was quantified. **C** Metastatic foci were identified based on imaging results, with the blue arrow indicating normal lymph nodes and the red arrow indicating metastatic lymph nodes. **D** H&E staining of metastatic foci and the incidence of metastasis per group

are shown. The migratory (**E**) and invasive (**F**) abilities of *F. nucleatum*-treated HNSCC cells were examined by Transwell assays (*n* = 6). **G** Cancer-endothelium adhesion assays were performed to examine the intercellular adhesion abilities of *F. nucleatum*-treated HNSCC cells (*n* = 6). **H** Transendothelial migration assays were performed to examine the transendothelial migration abilities of *F. nucleatum*-treated HNSCC cells (*n* = 6). The control (Ctrl) group refers to untreated cells. Bars indicate median with interquartile range (A, B), and mean with SD (E-H). ns, nonsignificant; $^{*}P < 0.05$, $^{**}P < 0.01$, $^{***}P < 0.001$, $^{****}P < 0.0001$.

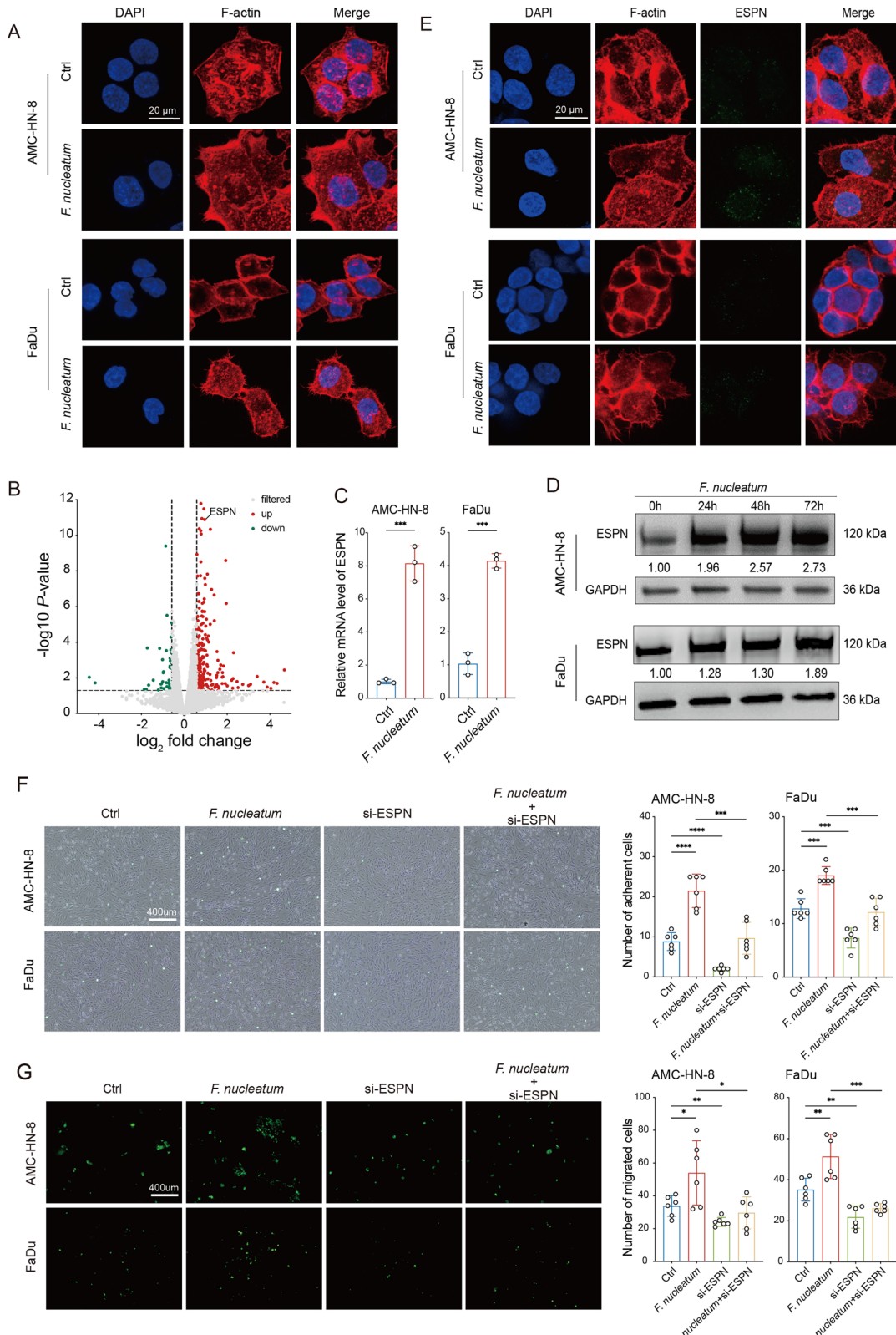

**Fig. 3 | *F. nucleatum* promotes HNSCC cells extravasation by upregulating ESPN protein. A** Representative confocal immunofluorescence images showing F-actin (red) in HNSCC cells. **B** Volcano plot showing differentially expressed genes (fold change >1.5 or <0.5, *P* < 0.05) in *F. nucleatum*-treated AMC-HN-8 cells identified by RNA sequencing. The red dots indicate the upregulated genes, while the green dots indicate the downregulated genes. RT-qPCR and Western blot analysis showing effect of *F. nucleatum* on ESPN mRNA (**C**) and protein (**D**) levels (*n* = 3). **E** Confocal immunofluorescence images showing F-actin (red) and ESPN (green) in *F. nucleatum*-treated HNSCC cells. **F** Cancer-endothelium adhesion assay showing the adhesion abilities of *F. nucleatum*-treated HNSCC cells pretreated with ESPN siRNA (*n* = 6). **G** Transendothelial migration assay showing the migration abilities of *F. nucleatum*-treated HNSCC cells pretreated with ESPN siRNA (*n* = 6). Ctrl, control group. Data are represented as mean with SD. *P < 0.05, **P < 0.01, ***P < 0.001, ****P < 0.0001.

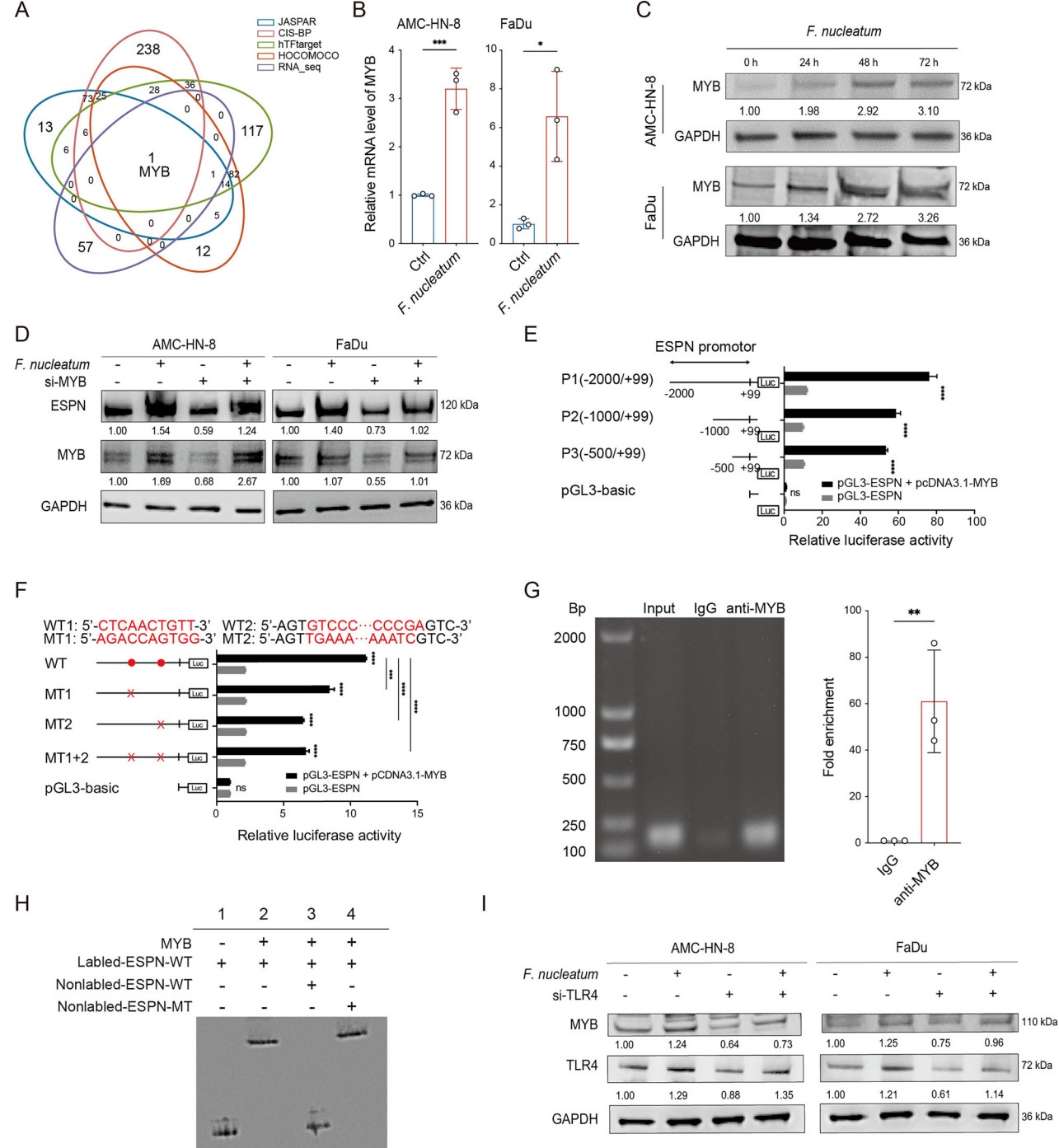

**Fig. 4 | *F. nucleatum* upregulates MYB to elicit ESPN transcription. A** Venn diagram showing the overlap between five datasets. Transcription factors predicted to bind the ESPN promoter from four databases (JASPAR, CIS-BP, hTFtarget, and HOCOMOCO) intersected with differentially expressed transcription factors from RNA sequencing. RT-qPCR and Western blot analysis showing effect of *F. nucleatum* on MYB mRNA (**B**) and protein (**C**) levels (*n* = 3). **D** Western blot analysis showing effect of MYB siRNA on ESPN expression in HNSCC cells. **E** Relative luciferase activities of reporters containing different fragments of the ESPN promoter. **F** Relative luciferase activities of reporters containing wild-type (WT) or mutated (MT) sequences of the ESPN promoter. Red letters represent putative or mutated MYB binding sequences. **G** ChIP-qPCR showing the binding of MYB to the ESPN promoter region (*n* = 3). **H** EMSA analysis of the interaction between MYB and the wild-type or mutated ESPN promoter in AMC-HN-8 cells (*n* = 3). **I** Western blot analysis showing effect of TLR4 siRNA on MYB expression in HNSCC cells. Ctrl, control group. Bars represent mean with SD. *$P < 0.05$, **$P < 0.01$, ***$P < 0.001$, ****$P < 0.0001$.

*nucleatum* is more common in metastatic than non-metastatic tissues[35,36] and is consistently present in both primary and metastatic tumors[14]. Extravasation is a pivotal phase in the process of cancer metastasis, requiring attachment of tumor cells to ECs and subsequent transendothelial migration[37,38]. We found that *F. nucleatum* enhances the adhesion to ECs and transendothelial migration abilities of HNSCC cells. Similarly, previous

research has shown that *F. nucleatum* upregulates ICAM1 expression, increasing CRC cell adhesion to ECs and promoting metastasis[39].

During extravasation, protrusive structures called filopodia form, allowing tumor cells to adhere to the endothelial surface, facilitating attachment and movement along the vessel wall[40]. We found that *F. nucleatum* induced filopodia extension in HNSCC cells, with transcriptome

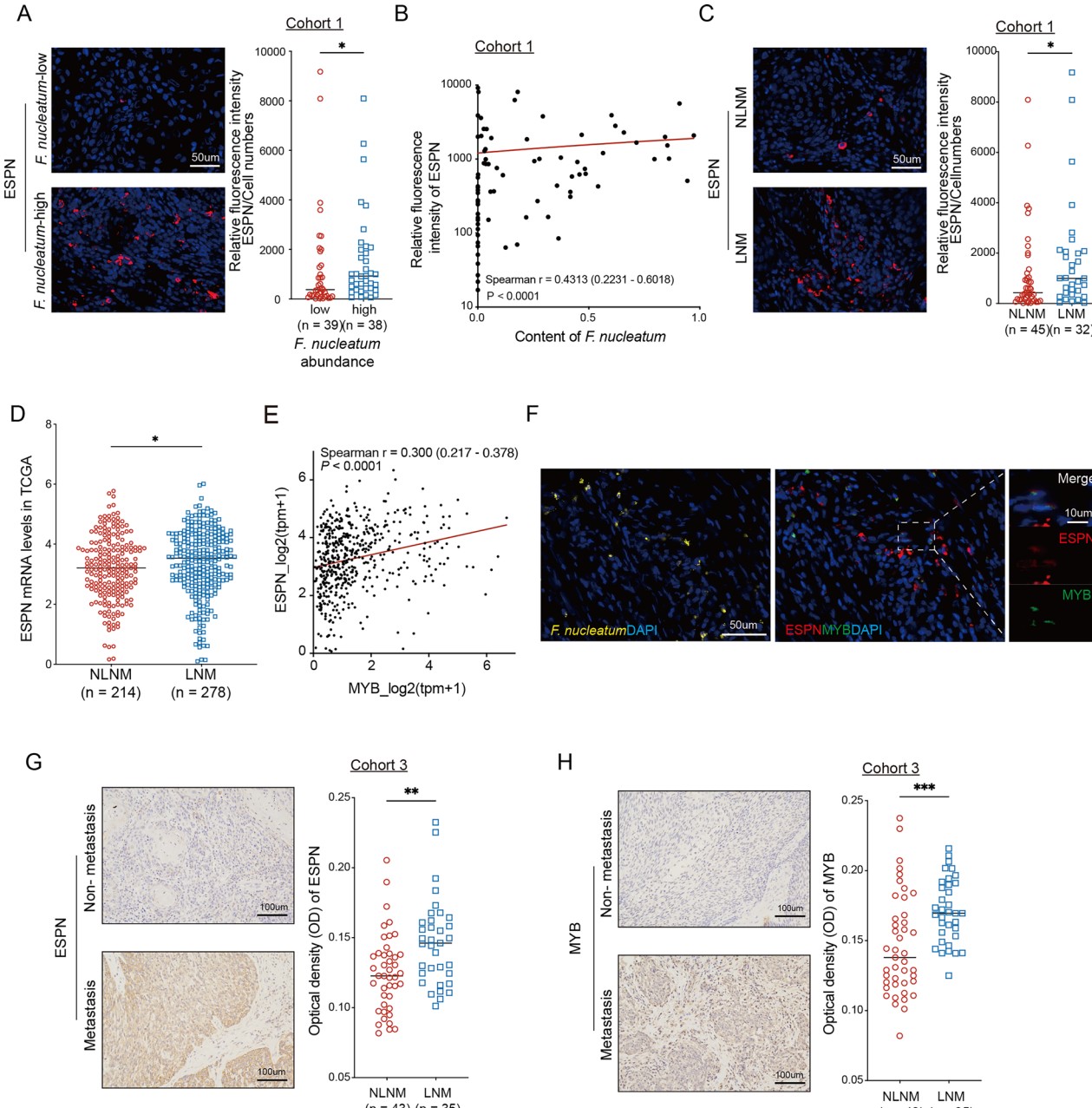

**Fig. 5 | ESPN and MYB levels are linked to *F. nucleatum* abundance and elevated in patients with HNSCC exhibiting LNM. A** Quantitative analysis of ESPN in Cohort 1 based on *F. nucleatum* abundance. **B** Correlation analysis between ESPN levels and *F. nucleatum* abundance in Cohort 1. **C** Representative immunofluorescence images and ESPN levels in tumor tissues with or without LNM in Cohort 1. **D** ESPN mRNA levels in tumor tissues with or without LNM in the TCGA-HNSC cohort (*n* = 492, excluding patients with unknown nodal status, *n* = 28). **E** Correlation between ESPN and MYB mRNA levels in the TCGA-HNSC cohort. **F** Representative immunofluorescence images of successive sections from HNSCC tissues showing the localization of *F. nucleatum* (yellow), ESPN (red), and MYB (green). Representative immunohistochemical images and ESPN (**G**) and MYB (**H**) levels in tumor tissues with or without LNM in Cohort 3. Bars indicate the median. *$P < 0.05$, **$P < 0.01$, ***$P < 0.001$.

sequencing revealing increased ESPN expression in response to *F. nucleatum* exposure. Inhibition of ESPN expression reversed *F. nucleatum*-induced enhancements in cell migration, invasion, endothelial adhesion, and transendothelial migration. We further observed that higher ESPN levels were significantly associated with LNM. As an actin-binding protein, ESPN plays a pivotal role in actin filament assembly and filopodia extension through its WH2 domain, which interacts with actin monomers[20,29,41]. Previous studies have shown high ESPN expression in melanoma tissues, where its downregulation reduced lamellipodia formation and invasiveness[22]. Similarly, miR-612 was found to suppress ESPN expression, limiting melanoma cell growth, migration, and invasion[42]. Additionally,

among patients with esophageal squamous cell carcinoma (ESCC), elevated ESPN expression has been linked to poor prognosis, and its reduction inhibited ESCC cell growth[24]. It is the first study to identify the critical function of ESPN in mediating *F. nucleatum*-induced invasion and metastatic potential in HNSCC.

Further investigation revealed that *F. nucleatum* upregulated the transcription factor MYB, with high MYB expression correlating with LNM. MYB, also known as c-MYB, is a proto-oncogene that encodes a transcription factor and is located at chromosome 6q23.3[43]. Researches have demonstrated aberrant MYB expression in multiple cancer types. Elevated MYB expression has been observed in esophageal cancer and correlates with

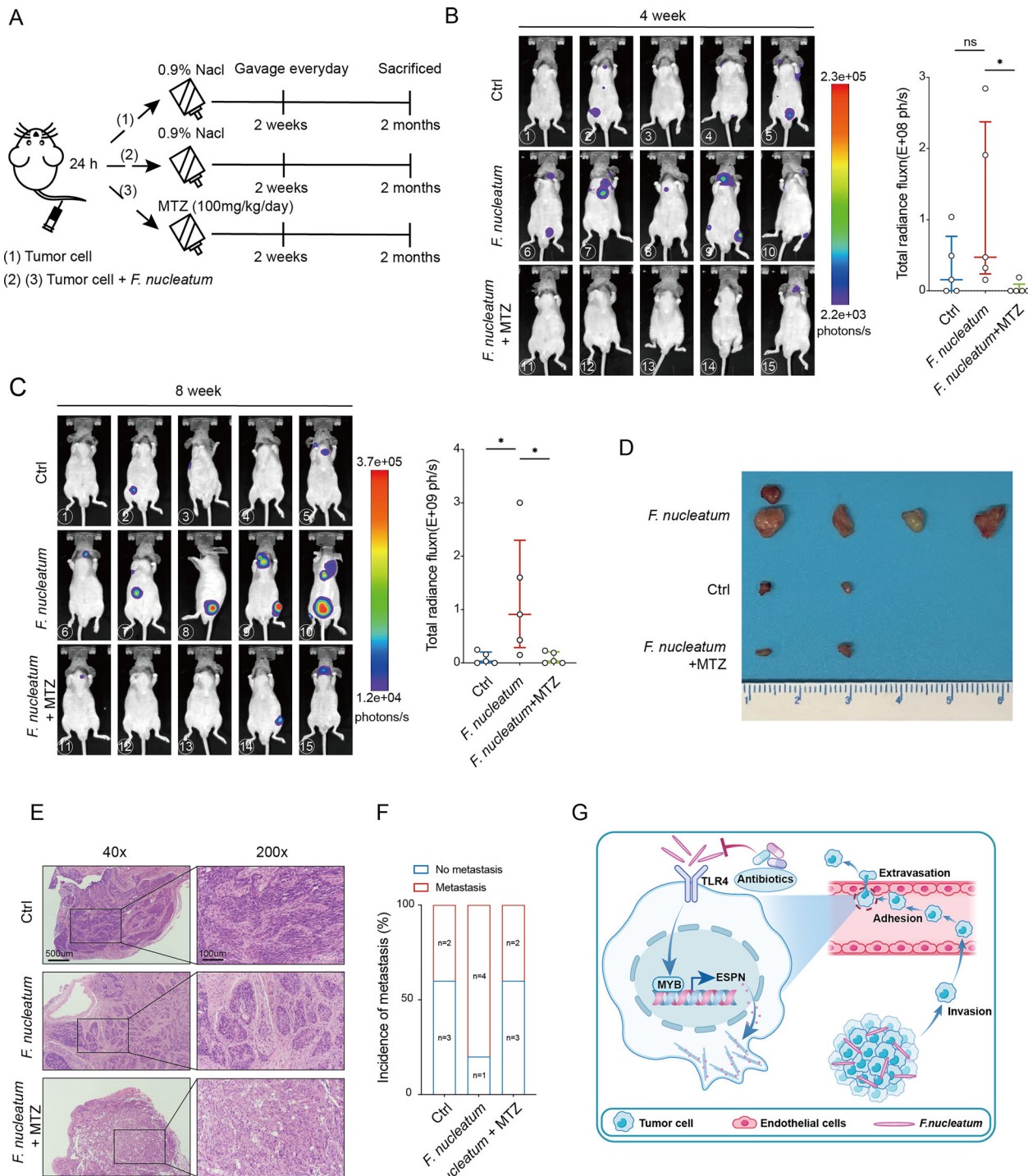

**Fig. 6 | Metronidazole treatment reduces *F. nucleatum*-mediated metastasis of HNSCC cells in vivo. A** Experimental scheme (*n* = 5 per group). AMC-HN-8 cells were intravenously injected after 24 h incubation with either *F. nucleatum* or PBS. MTZ (100 mg/kg/day) or saline was administered daily via oral gavage beginning the day after cell injection, continuing for two weeks. (**B** and **C**) Metastatic foci were detected using intravital imaging, and signal intensity was quantified (*n* = 5). **D** Representative gross images of metastatic foci. **E, F** H&E staining of metastatic foci and the incidence of metastasis per group are shown. **G** Schematic illustration of the mechanism by which *F. nucleatum* facilitates the extravasation and metastasis of HNSCC via TLR4/MYB/ESPN axis. *F. nucleatum* activates the transcription factor MYB through TLR4. MYB then binds to the ESPN promoter to initiate transcription, enhancing migration, invasion, endothelial adhesion, and transendothelial migration of HNSCC cells, ultimately increasing extravasation and metastatic potential. MTZ, metronidazole. Ctrl, control group. Bars represent the median with interquartile range. ns, nonsignificant; *P* < 0.05.

poor outcomes[44]. MYB is abnormally overexpressed in CRC tissues and linked to both LNM and poor prognosis[45]. Knockdown of the MYB gene was shown to suppress EMT in CRC cells[45]. In breast cancer, MYB enhances migration and invasion by regulating matrix metalloproteinase 9 (MMP9) and cathepsin D[46]. Moreover, MYB has been found to directly bind to the Axin2 promoter, activating the Wnt/β-catenin signaling pathway, which promotes invasion and metastasis in breast cancer[47]. In this study, *F. nucleatum*'s upregulation of ESPN was reversed by MYB silencing,

indicating that *F. nucleatum* stimulates ESPN expression via MYB. Mechanistic analysis confirmed that MYB directly binds to the ESPN promoter, regulating its transcription. Since *F. nucleatum* activates TLR4 signaling[31], it was demonstrated that *F. nucleatum* upregulates MYB in a TLR4-dependent manner. Thus, *F. nucleatum* activates the MYB/ESPN axis via TLR4 signaling, facilitating HNSCC cell interaction with ECs and promoting metastasis (Fig. 6G).

In vivo administration of MTZ following the injection of *F. nucleatum*-infected HNSCC cells led to a marked decrease in tumor cell metastatic potential. This finding aligns with previous studies, which demonstrated that MTZ can suppress *F. nucleatum*-associated tumor progression and metastasis[12,14]. In patients with CRC exhibiting high *F. nucleatum* abundance, reducing intestinal *F. nucleatum* levels through MTZ treatment was shown to enhance the efficacy of immunotherapy[48]. These results suggest significant therapeutic potential in targeting *F. nucleatum* for cancer treatment, offering a novel perspective: eliminating cancer-associated bacteria could play a critical role in cancer management. Understanding how microbiota influence cancer progression and treatment response presents considerable promise for identifying precise targets in metastasis therapy. MTZ is a broad-spectrum antibiotic that kills most anaerobic bacteria. This non-selective activity may adversely affect healthy commensal flora and potentially lead to secondary infections. Furthermore, the optimal treatment duration requires further investigation, as prolonged antibiotic use may drive antimicrobial resistance. Developing *Fusobacterium*-specific antimicrobials would represent a more targeted therapeutic approach. More preclinical researches are needed to facilitate clinical translation.

There are several limitations of this study. The detection of *F. nucleatum* using FISH in five paired tissue samples showed positive results in only three pairs. Although this rate is consistent with the known characteristics of intratumoral bacterial presence[49], the limited sample size constitutes a limitation of this study. Furthermore, the FISH method itself has a limitation: the spatial distribution of *F. nucleatum* in tissues is heterogeneous and variations in sampling location of FFPE sections may affect the detection results. Another limitation lies in the clinical samples. A priori sample size calculation was not performed for this study. A post-hoc power analysis revealed that the study had a power of 87.4% to detect the observed effect size for the primary endpoint, which indicates that the sample size was sufficient. However, further larger prospective studies are needed to validate these results. We did not evaluate p16 protein expression in tumor tissue, nor detected HPV DNA through PCR. Given the established prognostic impact of HPV status in head and neck cancers, its undetermined status in our cohorts may bring potential bias to the survival analysis. In addition, it is important to note that we did not investigate the potential involvement of other signaling cascades, such as NF-κB signaling pathway[50], outer membrane vesicles[51], or EMT programme[13], which have also been implicated in *F. nucleatum*-associated metastasis. Future study is needed to explore the potential regulatory crosstalk.

In conclusion, this study highlights that *F. nucleatum* fuels adhesion to vascular endothelium, transendothelial migration, and extravasation of HNSCC cells through the TLR4/MYB/ESPN axis, elucidating a key mechanism behind *F. nucleatum*-driven HNSCC metastasis. These observations provide novel insights for the treatment of HNSCC. Reducing *F. nucleatum* in tumor tissues using antibiotic therapy to inhibit metastasis offers a promising approach, though further clinical trials with larger cohorts are required to validate its potential role in cancer treatment.

## Materials and methods
### Clinical samples
This study was approved by the Ethics Committee of the Eye & ENT Hospital, Fudan University. All ethical regulations relevant to human research participants were followed. Tissue samples were collected from the Eye & ENT Hospital, Fudan University. For FISH analysis, paired FFPE tissue samples of pathologically confirmed primary tumors and metastatic lymph nodes were collected from five patients with HNSCC. A TMA was constructed using fresh HNSCC tumor and paired adjacent normal tissues, testing a total of 77 primary HNSCC tissues (Cohort 1, Table 1). Additionally, 104 FFPE HNSCC samples (Cohort 2, Table 1) were collected for genomic DNA extraction, while 78 FFPE HNSCC samples (Cohort 3, Table 1) were used for IHC analysis. All participants provided written informed consent. The mRNA expression profiling data for 520 primary tumors from the TCGA-HNSC project were obtained from TCGA database (https://portal.gdc.cancer.gov/). Gene expression profiles of GSE 65858 and GSE 117973 were obtained from the Gene Expression Omnibus (GEO) database (http://www.ncbi.nlm.nih.gov/geo/). *F. nucleatum* abundance data in the public database was obtained from the TCMA database (https://doi.org/10.7924/r4bk1j35s).

### Cell culture and bacteria strain
The human HNSCC cell line AMC-HN-8 (RRID: CVCL_5966) was generously provided by Professor S. Y. Kim, and the human HNSCC cell line FaDu (RRID: CVCL_1218) was purchased from the Cell Bank of Type Culture Collection of Chinese Academy of Sciences (CBTCCCAS). AMC-HN-8 and FaDu cells were cultured under the corresponding conditions. *F. nucleatum* strain ATCC 25586 was purchased from the American Type Culture Collection (ATCC). *F. nucleatum* was cultured at 37 °C under anaerobic conditions in a thioglycollate medium (without agar). Cells were co-incubated with *F. nucleatum* for 24 h at a multiplicity of infection (MOI) of 500:1 for the in vitro experiments.

### Metastasis mouse model
The male, 4-5 weeks old BALB/c-nu/nu mice were purchased from Shanghai Zhanluan Biotechnology Co. Ltd (China). The animal experiment protocol was approved by the Ethics Committee of the Eye & ENT Hospital, Fudan University. We have complied with all relevant ethical regulations for animal use. The maximal tumor size permitted by the ethics committee was 2 cm, and the maximal tumor size was not exceeded in this study. Mice were allocated to experimental and control groups using a simple randomization process. Following pretreatment with *F. nucleatum* (MOI = 200:1) or PBS for 24 h, $1.5 \times 10^6$ AMC-HN-8 cells stably expressing luciferase were injected via tail vein. For MTZ administration, mice received MTZ (100 mg/kg/day, 10 mg/mL suspension in physiological saline, #M1547, Sigma-Aldrich) by daily oral gavage starting the day after cell injection, continuing for two weeks. Experiments were conducted with 5 mice per group.

### In vivo fluorescence imaging
In vivo fluorescence imaging was conducted using the IndiGO platform (Berthold Technologies, Germany). Following intraperitoneal injection (10 μL/g) of D-luciferin (15 mg/mL suspension in PBS, #ST196, Beyotime), each mouse was placed in an anesthesia chamber with 2% isoflurane. Imaging was performed 10 min post-injection.

### Transwell assay
Transwell chambers (CORNING) were used in 24-well plates. Invasion assays were carried out using a 1:8 mixture of Matrigel (CORNING) and serum-free medium. A medium volume of 600 μL, supplemented with 10% FBS, was added to each well, and 100 μL of cell suspension (migration assay: $1 \times 10^5$ cells/mL; invasion assay: $1 \times 10^6$ cells/mL) was added to the chambers. Cells were fixed and stained after 72 h, followed by microscopic examination.

### Cancer endothelium adhesion assay
The 5-carboxyfluorescein diacetate N-succinimidyl ester (CFSE, #HY-D0056, MCE) was used to label tumor cells. Human umbilical vein endothelial cells (HUVECs) were cultured in a 6-well plate ($5 \times 10^5$ cells per well). Once a monolayer had formed, CFSE-labeled tumor cells ($7 \times 10^5$ cells per well) were introduced to the HUVEC-seeded wells. After 1 h incubation at 37 °C, cells were gently rinsed and examined using a fluorescence microscope.

## Transendothelial migration assay

The transendothelial migration assay was conducted using transwell chambers and 24-well plates. HUVECs were seeded into the transwell chambers ($5 \times 10^4$ cells). Following monolayer formation, CFSE-labeled tumor cells ($2 \times 10^4$ cells per well) were introduced into the chambers and 600 µL of medium containing 10% FBS was added to the wells. After 72 h, cells were fixed with 4% paraformaldehyde for 30 min and visualized under a fluorescence microscope.

## FISH

*F. nucleatum* was examined in TMA and FFPE sections, as previously described[17]. A 488-labeled universal bacterial probe (EUB338: 5'-GCTGCCTCCCGTAGGAGT-3')[52] and a Cy3-labeled probe specific for *F. nucleatum* (FUS664: 5'-CTTGTAGTTCCGC(C/T)TACCTC-3')[53] were used. The integrated fluorescence density of the EUB338 probe was set as 100% to represent the total bacterial biomass, and the *F. nucleatum* content was quantified as the ratio of the FUS664 probe's integrated fluorescence intensity to that of the EUB338 probe[54].

## RNA extraction, reverse transcription, and RNA sequencing

Total RNA from cell lines was extracted using the AG RNAex Pro Reagent (#AG21102, Accurate Biotechnology). Reverse transcription was performed with the Evo M-MLV RT Mix Kit with gDNA Clean for qPCR (#AG11728, Accurate Biology). RNA sequencing was performed following standard protocols[55].

## Real-time quantitative polymerase chain reaction (RT-qPCR)

The ABI 7500 Real-Time PCR System (Thermo Fisher, USA) was used to perform RT-qPCR on a 96-well optical PCR plate. A 10 µL reaction system was prepared with SYBR® Green Pro Taq HS premixed qPCR kit (#AG11718, Accurate Biology). The reference gene utilized in this study was the GAPDH primer. Relative gene expression was assessed using the $2^{-\Delta\Delta CT}$ method. The ESPN and MYB primers were as follow: ESPN forward: 5'-CCGCCAAAGGAGACTTCCC-3'; reverse: 5'-GTGGCACCGTTCTTGGTTT-3'; MYB forward: 5'-ATCTCCCGAATCGAACAGATGT-3'; reverse: 5'-TGCTTGGCAATAACAGACCAAC-3'.

## DNA extraction and *F. nucleatum* quantification

Genomic DNA (gDNA) from FFPE tissues was extracted according to the instructions of the GeneRead DNA FFPE Kit (#180134, Qiagen). *F. nucleatum* DNA was detected via RT-qPCR, following a previously described protocol[56]. The prostaglandin transporter (PGT) primer (forward: 5'-ATCCCCAAAGCACCTGGTTT-3', reverse: 5'-AGAGGCCAAGATAGTCCTGGTAA-3') was used as the reference primer, and the primers for *F. nucleatum* were as follows: forward: 5'-CAACCATTACTTTAACTCTA-3', reverse: 5'-GTTCAGTTGACTTTACAGAAGGAGATTATGTAAAAATC-3'. The relative abundance of *F. nucleatum* was calculated using the -ΔCt method.

## Western blot

After protein extraction and protein concentration quantification, the samples were subjected to heating at 70 °C for 10 min to induce protein denaturation. Following SDS-PAGE, the proteins were transferred onto nitrocellulose membranes. After blocking with 5% bovine serum albumin (BSA), the membranes were incubated with primary antibodies and secondary antibodies. The primary antibodies used were: ESPN (#A15908, Abclonal, RRID: AB_2763339), MYB (#17800-1-AP, Proteintech, RRID: AB_2148029), TLR4 (#19811-1-AP, Proteintech, RRID: AB_10638446), and GAPDH (#60004-1-Ig, Proteintech, RRID:AB_2107436).

## Immunofluorescence staining

For immunocytofluorescence, AMC-HN-8 and FaDu cells were fixed with 4% paraformaldehyde and then permeabilized using 0.5% Triton X-100. After blocking using 5% BSA, the cells were incubated with the primary antibody (ESPN, #NBP2-55817, Novus Biologicals, RRID: AB_3341008) overnight at 4 °C, followed by incubation with the secondary antibody (#ab150083, Abcam, RRID: AB_2714032). F-actin was stained with ActinRed 555 ReadyProbes Reagent (#R37112, ThermoFisher). Images were taken using a confocal microscope (Stellaris 5, Leica, Germany).

Double immunofluorescence staining was conducted using a TMA containing fresh HNSCC tissues. After deparaffinization and antigen retrieval, sections were treated with 5% BSA for blocking. Subsequently, sections were incubated overnight with anti-ESPN (#NBP2-55817, Novus Biologicals, RRID: AB_3341008) and anti-MYB (#ab45150, Abcam, RRID: AB_778878) primary antibodies. Secondary antibodies (#ab6939, #ab150077, Abcam, RRID: AB_955021, AB_2630356) were applied, and protein expression was analyzed by measuring integrated intensity across three microscopic fields (400 × magnification), normalized to the number of cells per field.

## Cell transfection

The siRNAs were transfected into cells in opti-MEM (Gibco). The following sequences of siRNAs were used: siESPN-1: 5'-GACAGGTCATCCTGAAGAA-3'; siESPN-2: 5'-GTGCCACAGTCTTGCATCT-3'; siESPN-3: 5'-GCTACGATGAGAGCAAGCT-3'; siMYB-1: 5'-GGAAAGTTATTGCCAATTA-3'; siMYB-2: 5'-GACACCCTCTCATCTAGTA-3'; siMYB-3: 5'-CCGAAACGTTGGTCTGTTA-3'; siTLR4-1: 5'-GTGCAATTTGACCATTGAA-3'; siTLR4-2: 5'-TGGTGAGTGTGACTATTGA-3'; siTLR4-3: 5'-CTACTACCTCGATGATATT-3'.

## IHC and H&E staining

IHC staining was conducted following established protocols[17]. Protein expression was quantified using mean optical density (MOD), with three distinct fields captured at 200x magnification for analysis. The primary antibodies used were ESPN (#A15908, Abclonal, RRID: AB_2763339) and MYB (#ab45150, Abcam, RRID: AB_778878). Hematoxylin and eosin (H&E) staining was applied following deparaffinization with xylene and ethanol.

## Dual-luciferase reporter gene assay

JASPAR (http://jaspar.genereg.net/), CIS-BP (http://cisbp.ccbr.utoronto.ca/), hTFtarget (http://bioinfo.life.hust.edu.cn/hTFtarget), and HOCOMOCO databases were utilized to predict transcription factors known to bind to the promoter region of ESPN. The plasmid construction sequences are provided in the Supplementary Table S3. The ESPN promoter sequence was cloned into the pGL3-basic plasmid, while the MYB sequence was inserted into the pcDNA3.1 plasmid. The pcDNA3.1-MYB plasmid was used to transfect HEK-293 cells along with either the mutant or wild-type pGL3-basic-ESPN plasmid. The activities of Renilla and firefly luciferase were measured and normalized after 48 h.

## ChIP

ChIP was carried out following the protocol of the Pierce Agarose ChIP kit (#26156, ThermoFisher). The ChIP was divided into three groups: positive control (input), negative control (IgG), and experimental (Anti-MYB). The antibody used for ChIP was MYB (#17800-1-AP, Proteintech, RRID: AB_2148029). Purified DNA was quantified via RT-qPCR using the following primers: Forward: 5'-CGACAATCACTCCCCATCACCTCT-3'; Reverse: 5'-GAGCCGTCCAGGCACAGACAAA-3'. Fold enrichment was calculated to analyze RT-qPCR results using the formula: $2^{[(CT\ IgG\ -\ CT\ input)\ -\ (CT\ anti-MYB\ -\ CT\ input)]}$.

## EMSA

NE-PER Nuclear and Cytoplasmic Extraction Kit (#78833, ThermoFisher) was used to carry out nuclear protein extraction. EMSA was conducted according to the protocol of LightShift® Chemiluminescent EMSA Kit (#20148, ThermoFisher). The probes used for EMSA are shown in the Supplementary Table S4.

## Statistics and reproducibility

All experiments were performed at least three times. The data analysis was performed blind to the group allocation. Statistical analyses were performed using GraphPad Prism version 9.0. PASS 15.0.5 was used to conduct a post-hoc analysis. In the case of normally distributed variables, Student's $t$-tests were used, while in the case of non-normally distributed variables, Mann-Whitney U tests were used. Survival curves were generated using Kaplan–Meier analysis, with group differences assessed by the log-rank test. The mean ± standard deviation (SD) or the median was used to express the data. A $P$-value of less than 0.05 was considered statistically significant.

## Ethics

All experiments and methods were performed in accordance with relevant guidelines and regulations. The study was approved by the Ethics Committee of the Eye & ENT Hospital, Fudan University, China (approval number: 2022076). All participants signed informed consent.

## Reporting summary

Further information on research design is available in the Nature Portfolio Reporting Summary linked to this article.

## Data availability

The source data can be found in Supplementary Data 1. The transcriptome sequencing data is available in the NCBI GEO repository (https://www.ncbi.nlm.nih.gov/geo/) under accession number GSE236237 (DOI: 10.1099/mgen.0.001221). All other raw data generated in this study are available from the corresponding author upon reasonable request.

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

## Acknowledgements

This study was supported by the National Natural Science Foundation of China (81502343, 81972529, 82203727), and Shanghai Sailing Program (22YF1405700).

## Author contributions

H.G., C.H., and L.Z. designed and supervised the study. X.Y. wrote the manuscript. X.Y. and H.H. performed the experiments and analyzed the data. H.L., Q.H., Y.S., and Z.W. collected the data. M.Z. and L.T. assisted with sample collection and provided guidance.

## Competing interests

The authors declare no competing interests.
