## [Transparent Peer Review file · Communications Biology]

Fusobacterium nucleatum promotes tumor extravasation and metastasis in head and neck cancer via TLR4/MYB/ESPN axis

Corresponding Author: Dr Chi-Yao Hsueh

Version 0:

Reviewer comments:

Reviewer #1

(Remarks to the Author)

The study by Yuan et alii shows that *Fusobacterium nucleatum* is more abundant in head and neck squamous cell carcinoma (HNSCC) patients with lymph node metastasis and is linked to worse disease-free survival. The bacteria invade both primary and metastatic tumors, promote tumor cell adhesion and migration by increasing ESPN expression, and do so via TLR4–MYB signaling. In animal models, metronidazole treatment reduced metastasis, suggesting *F. nucleatum* could be a therapeutic target in metastatic HNSCC.

While this is a very interesting topic with promising findings there are some major considerations before publication to translate the findings in the clinical setting. This includes:

1) Abstract: indicate "ESPN"

2) Intro: Line 72, please give examples of cancers, especially carcinomas (SCC?). In General, Please respect the fact of HPV +/- cancers and the various risk factors for HNSCC over the globe and its potential influence in the microbiome of HNSCC.

3) results: please indicate, whether HNSCC samples were HPV +/-

please indicate more information about cohort 1 and 2 in this section (refer to table in M+M). For the reader it is difficult to understand how the cohorts were built.

The sentence "H&E staining confirmed that *F. nucleatum* increased the incidence of metastasis" should be rewritten because its current meaning is that the bacteria were detected by HE (and not, as intended, HE showed more metastases in the FB+ mice).

4) discussion: please give more information about MYB in general and its interaction with metronidazole as this is a therapeutic approach. Can this be translated into patient care one day? And if so, how can the treatment be monitored?

Reviewer #2

(Remarks to the Author)

General Assessment

This manuscript entitled "Fusobacterium nucleatum promotes tumor extravasation and metastasis in head and neck squamous cell carcinoma via TLR4/MYB/ESPN axis" investigates the role of *F. nucleatum* in promoting HNSCC metastasis. The authors provide evidence from patient cohorts, in vitro mechanistic studies, and in vivo mouse models. The topic is timely and of high interest, as intratumoral microbiota is emerging as an important factor in cancer progression. The mechanistic focus on the TLR4/MYB/ESPN axis is novel and clinically relevant. Overall, the study has potential for publication in Communications Biology, but several important issues should be addressed prior to acceptance.

Major Comments

1. Sample size and statistical power

· The study analyzes Cohort 1 (n=77), Cohort 2 (n=104), and Cohort 3 (n=78). While these numbers are not negligible, they remain modest for survival and stratification analyses in HNSCC, particularly given the heterogeneity of tumor sites, clinical

stages, and treatment modalities.

- For example, subgroup analyses (LNM vs. NLNM, stage I–II vs. III–IV) are based on relatively small case numbers (e.g., only 9 patients in stage I–II in Cohort 1), which raises concerns about robustness.
- The manuscript would be strengthened by validating findings in an additional independent cohort (≥ 150 cases) or by performing cross-validation using publicly available datasets beyond TCGA-HNSC.

2. Sample size justification

- The authors do not explain how the sample size was determined. A priori power calculations (e.g., expected hazard ratio for DFS, significance level, and power) should be included or at least discussed. Without such justification, it is unclear whether the study is sufficiently powered to draw definitive conclusions.

3. Mechanistic scope

- While the TLR4/MYB/ESPN pathway is well documented here, the involvement of other pathways (e.g., NF- κ B, EMT-related transcription factors) is not examined. Given previous reports linking F. nucleatum to multiple signaling cascades, the authors should acknowledge this limitation.

4. Therapeutic implications of metronidazole

- The use of metronidazole to suppress F. nucleatum-driven metastasis is intriguing. However, the authors should discuss limitations such as (i) potential antibiotic resistance, (ii) non-specific effects on commensal microbiota, and (iii) translational challenges in clinical practice.

Minor Comments

- Clarify the dose and duration rationale for metronidazole treatment in mice. Was it based on previous CRC studies or pilot experiments?
- Figures 2 and 6 require improved labeling for clarity (e.g., arrows, scale bars, sample numbers).
- The English language should be revised for grammar and flow to improve readability.

Recommendation

Major revision. The study is of high potential interest, but issues regarding sample size justification, robustness of survival analysis, and discussion of limitations should be addressed before the manuscript can be considered for acceptance.

Version 1:

Reviewer comments:

Reviewer #1

(Remarks to the Author)

all my points have been addressed satisfactorily. However, due to the limited sample size, the general findings are with regard to the heterogeneity of HNSCC questionable and I wonder, whether such a renomme journal should publish the data.

Dear Editor and Reviewers

Thank you for your letter and comments concerning our manuscript entitled “*Fusobacterium nucleatum* promotes tumor extravasation and metastasis in head and neck squamous cell carcinoma via TLR4/MYB/ESPN axis” (COMMSBIO-25-6113). We sincerely appreciate your decision and constructive comments on the manuscript. We have carefully considered the suggestions of reviewers and have revised the manuscript. Revision notes, point-to-point, are given as follows:

Reviewers' comments:

Reviewer 1: The study by Yuan et alii shows that *Fusobacterium nucleatum* is more abundant in head and neck squamous cell carcinoma (HNSCC) patients with lymph node metastasis and is linked to worse disease-free survival. The bacteria invade both primary and metastatic tumors, promote tumor cell adhesion and migration by increasing ESPN expression, and do so via TLR4–MYB signaling. In animal models, metronidazole treatment reduced metastasis, suggesting *F. nucleatum* could be a therapeutic target in metastatic HNSCC. While this is a very interesting topic with promising findings there are some major considerations before publication to translate the findings in the clinical setting.

Comments:

1. Abstract: indicate "ESPN"

Author response: We have added an indication of ESPN in Abstract (Page 2, Line 24-26). The corresponding text now reads:

“*F. nucleatum* induced adhesion to endothelial cells and facilitated transendothelial migration via upregulating ESPN expression in HNSCC cells, which is an actin-binding protein.”

2. Intro:

Point 1: Line 72, please give examples of cancers, especially carcinomas (SCC?).

Point 2: In general, please respect the fact of HPV +/- cancers and the various risk

factors for HNSCC over the globe and its potential influence in the microbiome of HNSCC.

Author response: We thank the reviewer for this suggestion to improve the clarity of our introduction. We have added the relevant content in the Introduction (Page 3, Line 48-61 and Page 4-5, Line 86-97). The corresponding text now reads:

Point 1: “ESPN is crucial for cytoskeletal remodeling, especially in the formation and extension of cellular protrusions like filopodia and stereocilia, highlighting its role in facilitating tumor migration and invasion in cancers. In melanoma, ESPN is highly expressed in melanoma and knockdown of ESPN expression resulted in reduced lamellipodia formation as well as decreased activities of FAK and Rho GTPases in melanoma cells. In a recent research, ESPN was reported to promote osteochondroma metastasis. ESPN activated the AKT/mTOR pathway via direct interaction with and phosphorylation of PI3K, which enhanced ZEB1 expression to initiate EMT. High expression of ESPN is associated to poor prognosis in esophageal squamous cell carcinoma. However, the role of ESPN in promoting metastasis in squamous cell carcinoma, particularly in HNSCC, remains unclear.”

Point 2: “The HNSCC tumorigenesis results from a complex interplay between genetic factors and environmental exposures, showing considerable levels of heterogeneity. In addition to the primary risk factors for HNSCC—tobacco use, alcohol consumption, and human papillomavirus (HPV) infection—emerging attention has been directed toward the potential role of microbiota in HNSCC development. A large study revealed that heavy drinkers exhibit significant changes in their oral microbial structure, showing the influence of alcohol on the oral microbiome. Notably, the abundance of specific bacteria increases under the influence of alcohol, and their metabolites—which may serve as a source of carcinogenic acetaldehyde—could play a role in mediating the carcinogenic effects of alcohol consumption. HPV-positive HNSCC exhibits alterations in the abundance of specific bacteria and on the other hand, certain oral microbiota structure may increase host susceptibility to HPV infection and potentiate its cancer-promoting effects. The

human microbiota has gained increasing recognition as a key player in cancer development and progression.”

3. Results:

Point 1: Please indicate, whether HNSCC samples were HPV +/-

Point 2: Please indicate more information about cohort 1 and 2 in this section (refer to table in M+M). For the read it is difficult to understand how the cohorts were built.

Point 3: The sentence "H&E staining confirmed that *F. nucleatum* increased the incidence of metastasis" should be rewritten because its current meaning is that the bacteria were detected by HE (and not, as intended, HE showed more metastasis in the FB+ mice).

Author response:

Point 1: We did not examine HPV status of tumor samples. We have discussed this limitation in the discussion (Page 28, Line 510-513). The corresponding text now reads:

“We did not evaluate p16 protein expression in tumor tissue, nor detected HPV DNA through PCR. Given the established prognostic impact of HPV status in head and neck cancers, its undetermined status in our cohorts may bring potential bias to the survival analysis.”

Point 2: We thank the reviewer for this suggestion to improve the clarity of our cohort description. As suggested, we have now revised the corresponding section in the Results (Page 5-6, Lines 115-120) to provide a clearer summary of how Cohorts 1 and 2 were constructed. The corresponding text now reads:

“To further elucidate the association between *F. nucleatum* and LNM, we constructed a tissue microarray (TMA) using fresh HNSCC tumor tissues (Cohort 1, n = 77) and collected formalin-fixed paraffin-embedded (FFPE) HNSCC samples (Cohort 2 n = 104). Clinical data are detailed in Table 1. *F. nucleatum* levels in HNSCC tissues (Cohort 1) were quantified using FISH.”

Point 3: We sincerely appreciate the reviewer for pointing out this lack of clarity in our phrasing. We have rewritten the sentence in the Results section (Page 10, Line 183-184) to accurately reflect our findings: “H&E staining confirmed that the incidence of metastasis increased in *F. nucleatum*-treated group”

4. Discussion: Please give more information about MYB in general and its interaction with metronidazole as this is a therapeutical approach. Can this be translated into patient care one day and if so, how can the treatment be monitored?

Author response: We have added information about MYB in Discussion section (Page 27, Line 463-467): “MYB, also known as c-MYB, is a proto-oncogene that encodes a transcription factor and is located at chromosome 6q23.3. Researches have demonstrated aberrant MYB expression in multiple cancer types. Elevated MYB expression has been observed in esophageal cancer and correlates with poor outcomes.”

Although a direct interaction between MYB and metronidazole has not been documented, it is plausible that metronidazole may influence MYB expression by remodeling tumor microenvironment, such as by altering the composition of the microbial community and its metabolites. In our study, *F. nucleatum* can promote MYB expression, while metronidazole targeting *F. nucleatum* can block this pathway. The limitation of clinical translation of metronidazole has been addressed in the Discussion section (Page 28, Line 492-498), while its clinical applicability remains to be further explored. The corresponding text now reads:

“MTZ is a broad-spectrum antibiotic that kills most anaerobic bacteria. This non-selective activity may adversely affect healthy commensal flora and potentially lead to secondary infections. Furthermore, the optimal treatment duration requires further investigation, as prolonged antibiotic use may drive antimicrobial resistance.

Developing *Fusobacterium*-specific antimicrobials would represent a more targeted therapeutic approach. More preclinical researches are needed to facilitate clinical translation.”

Reviewer #2: This manuscript entitled “*Fusobacterium nucleatum* promotes tumor extravasation and metastasis in head and neck squamous cell carcinoma via TLR4/MYB/ESPN axis” investigates the role of *F. nucleatum* in promoting HNSCC metastasis. The authors provide evidence from patient cohorts, in vitro mechanistic studies, and in vivo mouse models. The topic is timely and of high interest, as intratumoral microbiota is emerging as an important factor in cancer progression. The mechanistic focus on the TLR4/MYB/ESPN axis is novel and clinically relevant. Overall, the study has potential for publication in Communications Biology, but several important issues should be addressed prior to acceptance.

Major Comments

1. Sample size and statistical power

- The study analyzes Cohort 1 (n=77), Cohort 2 (n=104), and Cohort 3 (n=78). While these numbers are not negligible, they remain modest for survival and stratification analyses in HNSCC, particularly given the heterogeneity of tumor sites, clinical stages, and treatment modalities.
- For example, subgroup analyses (LNM vs. NLNM, stage I–II vs. III–IV) are based on relatively small case numbers (e.g., only 9 patients in stage I–II in Cohort 1), which raises concerns about robustness.
- The manuscript would be strengthened by validating findings in an additional independent cohort (≥ 150 cases) or by performing cross-validation using publicly available datasets beyond TCGA-HNSC.

Author response: We sincerely thank the reviewer for these comments regarding the sample size and statistical power. Cohort 1 in this study was constructed using a tissue microarray (TMA). To ensure the successful completion of the TMA, priority was given to enrolling advanced-stage tumors from which larger fresh tissue specimens were more readily obtainable, a process that may have introduced selection bias. We recognized the limited number of Stage I/II cases in this cohort and therefore independently assembled another cohort (Cohort 2) consisting of 104 FFPE samples. Issues regarding statistical power have been addressed in our response to Comment 2.

To confirm the generality of our findings, we validated these results using public cohort from the TCMA and GEO databases¹. We observed a trend ($P = 0.0972$) towards higher *F. nucleatum* abundance in advanced-stage tumors among HNSCC samples where *F. nucleatum* was detectable. The TCMA cohort did not reveal a significant difference in *F. nucleatum* abundance regarding LNM. In contrast, a study regarding oral carcinoma reported a lower incidence of LNM in *F. nucleatum*-positive tumor tissues². These findings are likely attributable to the heterogeneity of head and neck cancer. Future investigations with larger sample sizes are needed to elucidate the relationship between *F. nucleatum* and LNM. We have added this discussion into the revised manuscript (Page 25, Line 428-435). Furthermore, we established the Cohort 3 specifically to validate the association between ESPN and LNM and this association was validated using the GEO database. These results have been included in the Supplementary Figures (Supplementary Figure S1) and incorporated into the Results section (Page 6, Line 126-129; Page 21-22, Line 362-364). The corresponding text now reads:

Results section:

“Similar results by TCMA cohort are also illustrated in Supplementary Fig. S1A.

However, the association between *F. nucleatum* abundance and LNM were not found in the TCMA cohort (Supplementary Fig. S1B).”

“Furthermore, the expression level of ESPN in HNSCC samples was validated using the GEO database (Supplementary Fig. S1C and D).”

Figure S1. Validation of the association of *F. nucleatum* abundance and ESPN expression with lymph node metastasis using public databases. (A) Relative abundance of *F. nucleatum* in patients with HNSCC with or without lymph node

metastasis (LNM) in TCMA cohort (n = 126, samples without detectable *F. nucleatum* were excluded). (B) Relative abundance of *F. nucleatum* in patients with HNSCC across different TNM stages in TCMA cohort. (C) ESPN mRNA levels in tumor tissues with or without LNM in GSE65858 cohort (n = 270). (D) ESPN mRNA levels in tumor tissues with or without LNM in GSE6117973 cohort (n = 77). Bars indicate mean with SD (A, B, D), and median with interquartile range (C). ns, nonsignificant; * $P < 0.05$. LNM, lymph node metastasis. NLNM, no lymph node metastasis.

Discussion section:

“However, The TCMA cohort did not reveal a significant difference in *F. nucleatum* abundance regarding LNM. In contrast, a study regarding oral carcinoma reported a lower incidence of LNM in *F. nucleatum*-positive tumor tissues. The inherent heterogeneity of HNSCC must be considered. Tumors originating from different anatomical sites exhibit distinct molecular characterization and tumor behaviors. The oncogenic effects of *F. nucleatum* may demonstrate significant heterogeneity depending on the specific anatomical site and molecular subtype context in which it acts.”

Reference:

1. Dohlman, A. B. *et al.* The cancer microbiome atlas: a pan-cancer comparative analysis to distinguish tissue-resident microbiota from contaminants. *Cell Host Microbe* **29**, 281-298.e5 (2021).
2. Neuzillet, C. *et al.* Prognostic value of intratumoral *Fusobacterium nucleatum* and association with immune-related gene expression in oral squamous cell carcinoma patients. *Sci Rep* **11**, 7870 (2021).

2. Sample size justification

The authors do not explain how the sample size was determined. A priori power calculations (e.g., expected hazard ratio for DFS, significance level, and power) should be included or at least discussed. Without such justification, it is unclear

whether the study is sufficiently powered to draw definitive conclusions.

Author response:

We thank the reviewer for raising this critical methodological point. We acknowledge that a priori sample size calculation was not performed for this study. To address the reviewer's concern regarding statistical power, we have performed a post-hoc power analysis. Based on the observed hazard ratio of 2.207, our actual sample size of 104, and a two-sided α of 0.05, the achieved power for detecting the observed effect was calculated to be 0.874 using PASS 15 (Page 6, Line 132-135). The limitation of this approach has been addressed in the Discussion section (Page 28, Line 506-510). The corresponding text now reads:

Results section:

“Given an a priori sample size calculation was not performed, we performed a post-hoc power analysis. Based on the observed hazard ratio of 2.207, actual sample size of 104, and a two-sided α of 0.05, the achieved power for detecting the observed effect was calculated to be 0.874.”

Discussion section:

“Another limitation lies in the clinical samples. A priori sample size calculation was not performed for this study. A post-hoc power analysis revealed that the study had a power of 87.4% to detect the observed effect size for the primary endpoint, which indicates that the sample size was sufficient. However, further larger prospective studies are needed to validate these results.”

3.Mechanistic scope

While the TLR4/MYB/ESPN pathway is well documented here, the involvement of other pathways (e.g., NF- κ B, EMT-related transcription factors) is not examined. Given previous reports linking *F. nucleatum* to multiple signaling cascades, the authors should acknowledge this limitation.

Author response:

We completely agree with the reviewer. As suggested, we have now acknowledged

this limitation in the revised manuscript (Page 28-29, Line 513-518): “In addition, it is important to note that we did not investigate the potential involvement of other signaling cascades, such as NF- κ B signaling pathway, outer membrane vesicles, or EMT programme, which have also been implicated in *F. nucleatum*-associated metastasis. Future study is needed to explore the potential regulatory crosstalk.”

4. Therapeutic implications of metronidazole

The use of metronidazole to suppress *F. nucleatum*-driven metastasis is intriguing. However, the authors should discuss limitations such as (i) potential antibiotic resistance, (ii) non-specific effects on commensal microbiota, and (iii) translational challenges in clinical practice.

Author response:

We thank the reviewer for this insightful and constructive comment. As suggested, we have added the related discussion into revised manuscript (Page 28, Line 492-498). The corresponding text now reads: “MTZ is a broad-spectrum antibiotic that kills most anaerobic bacteria. This non-selective activity may adversely affect healthy commensal flora and potentially lead to secondary infections. Furthermore, the optimal treatment duration requires further investigation, as prolonged antibiotic use may drive antimicrobial resistance. Developing *Fusobacterium*-specific antimicrobials would represent a more targeted therapeutic approach. More preclinical researches are needed to facilitate clinical translation.”

Minor Comments

• Clarify the dose and duration rationale for metronidazole treatment in mice. Was it based on previous CRC studies or pilot experiments?

Author response: It was based on a previous CRC studie¹.

Reference:

1. Dong, J. *et al.* Oral microbiota affects the efficacy and prognosis of radiotherapy

for colorectal cancer in mouse models. *Cell Rep* **37**, 109886 (2021).

• Figures 2 and 6 require improved labeling for clarity (e.g., arrows, scale bars, sample numbers).

Author response: We have improved the clarity of the arrows, numerical labels, and error bars in Figures 2 and 6.

Figure 2.

Figure 6

• The English language should be revised for grammar and flow to improve readability.

Author response: We have polished the manuscript for grammar and flow.

Dear Reviewer

Thank you for your comments concerning our manuscript entitled “*Fusobacterium nucleatum* promotes tumor extravasation and metastasis in head and neck cancer via TLR4/MYB/ESPN axis” (COMMSBIO-25-6113A). We have carefully considered the suggestions. The point-by-point response is as follows:

Reviewer's comments:

Reviewer 1: all my points have been addressed satisfactorily. However, due to the limited sample size, the general findings are with regard to the heterogeneity of HNSCC questionable and I wonder, whether such a renomme journal should publish the data.

Author response: We appreciate the point on the limited sample size. Head and neck cancers exhibit considerable heterogeneity in biologic behavior and therapeutic response. Tumors originating from different anatomical sites exhibit distinct molecular characterization. The oncogenic effects of *Fusobacterium nucleatum* may demonstrate significant heterogeneity depending on the specific anatomical site and molecular subtype context in which it acts. In our study, while the clinical sample size is limited, the primary objective was to perform an in-depth, mechanistic investigation into *Fusobacterium nucleatum*-driven HNSCC metastasis. Specifically, we demonstrated that *Fusobacterium nucleatum* activates TLR4/MYB/ESPN axis to promote tumor extravasation. Larger, well-stratified patient cohorts are needed to validate the clinical prevalence and subtype-specificity of this mechanism.